# An IoT-Enabled Platform for the Assessment of Physical and Mental Activities Utilizing Augmented Reality Exergaming

**DOI:** 10.3390/s22093181

**Published:** 2022-04-21

**Authors:** Dionysios Koulouris, Andreas Menychtas, Ilias Maglogiannis

**Affiliations:** 1BioAssist SA, 11524 Athens, Greece; denkoul@bioassist.gr (D.K.); amenychtas@bioassist.gr (A.M.); 2Department of Digital Systems, University of Piraeus, 80, M. Karaoli & A. Dimitriou St., 18534 Piraeus, Greece

**Keywords:** augmented reality, exergames, gamification, IoT, sensors, mobile platforms, wearables

## Abstract

Augmented reality (AR) and Internet of Things (IoT) are among the core technological elements of modern information systems and applications in which advanced features for user interactivity and monitoring are required. These technologies are continuously improving and are available nowadays in all popular programming environments and platforms, allowing for their wide adoption in many different business and research applications. In the fields of healthcare and assisted living, AR is extensively applied in the development of exergames, facilitating the implementation of innovative gamification techniques, while IoT can effectively support the users’ health monitoring aspects. In this work, we present a prototype platform for exergames that combines AR and IoT on commodity mobile devices for the development of serious games in the healthcare domain. The main objective of the solution was to promote the utilization of gamification techniques to boost the users’ physical activities and to assist the regular assessment of their health and cognitive statuses through challenges and quests in the virtual and real world. With the integration of sensors and wearable devices by design, the platform has the capability of real-time monitoring the users’ biosignals and activities during the game, collecting data for each session, which can be analyzed afterwards by healthcare professionals. The solution was validated in real world scenarios and the results were analyzed in order to further improve the performance and usability of the prototype.

## 1. Introduction

Advancements in the technological fields of augmented reality (AR) and the Internet of Things (IoT) contribute, to a great extent, to the creation of innovative solutions, with unique characteristics for user interactivity. AR and IoT may provide modern features to applications, allowing them to support complex usage scenarios and to effectively address the challenges and requirements for seamless user experiences. In the areas of healthcare and assisted living, a long-standing need is to promote the increase of users’ physical activity, which, in the modern way of sedentary lifestyles, has been considerably reduced. Moreover, in many cases, interventions and incentives for consistently following a care plan are also required.

AR technology refers to a real-time direct or indirect view of a physical real-world environment that has been enhanced/augmented by adding virtual computer-generated information to it [1]. In the video games industry, AR solutions [2] are quite common nowadays, and typically gameplay scenarios impel players to increase their physical activity. Such games are called exergames and encourage the players to perform exercises and movements that, under other conditions, they would not perform [3,4,5]. In order to achieve the goals of such serious games, the use of gamification techniques is mandatory [6], and through the different goals of a game, players are required to perform physical and mental challenges to improve, mainly in a mid- to long-term time frames, their physical conditions and lifestyles. Most interventions aiming for an increase in physical activity require the use, installation, and configuration of expensive hardware (e.g., consoles, external modules, cameras, sensors, etc.), processes that can be quite complex at times.

The motivation for this work is to create a prototype solution for serious exergame development, which will not only promote healthy lifestyles, but also support healthcare professionals in the assessment of user health conditions and in the detection of risks. In addition, the overall solution should follow a modular design and utilize commodity hardware in order to ensure simplicity, interoperability with other platforms, openness to integrate with other modern technologies, and low costs. The proposed AR game development platform integrates via designed IoT technologies, tools for advanced user interactivity, and effective assessment of users’ vital signs and activity. The prototype includes three main elements: (a) an AR-ready mobile framework that is based on [7], (b) an application for IoT wearables for tracking user activity and biosignals, and (c) a cloud platform with a web application endpoint for healthcare professionals to remotely configure the game aspects and analyze the results. This approach, on one hand, utilizes AR development principles [8] for advanced interactivity, and on the other hand, incorporates IoT technologies for biosignal and activity monitoring while playing.

The use of gamification and AR exergames on commodity devices and across hardware platforms, for providing gameplay location independence, ease of use, remote configuration, and health monitoring, involves a combination of methodologies and technologies, which is not present in the current state of research. This work proposes a solution that effectively addresses the above needs using the cutting edge features of IoT, AR for mobile devices, data federations, and cloud platforms. The major contribution of the proposed platform for exergame development is that, in order to play the game and perform the various operations, no extra hardware is required, eliminating the need for extra costs or location limitations due to equipment dependencies. It can be supported by a vast majority of mobile devices; thus, offering ease of access and use. The users–players are able to exploit the advanced interactivity features of AR, monitor their health measurements during the game (e.g., number of steps, heart rate, and trembling of their hand), and store the results. In parallel, the professional can adapt and reconfigure the gameplay aspects, according to the condition the user and the care plan, after analyzing the results of the game sessions and the monitored activity and biosignal data. The user interface delivers simplicity in a general user experience when displaying in-game real time metrics. This example demonstrates the benefits of developing AR mobile exergames and remote monitoring for healthcare use cases, and proves the positive impact on increasing the users’ physical activity and well-being. The results from the game show that users of different ages perform well while playing the game and it increase their physical activity in a pleasant and creative way, which keeps both their bodies and minds active.

The rest of the paper is structured according as follows: Section 2 analyzes the related research concerning AR, exergaming, and IoT in healthcare and assisted living scenarios, and presents details about the architecture, system design, and implementation of the prototype. Real-time game scenarios, performance, and medical results taken from the application in real-time conditions are presented in Section 3. Section 4 compares our work with related applications and technologies and also describes future extensions of this work.

## 2. Related Work

The design and the implementation of the concept and the prototype was based on the combination, in a modular exergame platform, of two different technological areas, AR and IoT. This solution both monitors users’ health statuses and also provides incentives for increasing users’ physical activities, highlighting the advantages of developing exergames on mobile devices using AR. Such interventions have been the core subjects of studies, especially in recent years [9] and demonstrate positive results [10,11], including children and elders [12]. Similar approaches implement human motion analysis methodologies and tools for exergame development, which are based on computer vision, deep learning technologies, and edge computing technologies [13]. Active video games in classrooms motivate students to conduct physical exercise [14], resulting in healthier lifestyles. Exergames can promote physical activity in the scope of personalised health [15] and are defined as a combination of exertion and video games [16]. Moreover, for elderly people, they aim at continuously mobilizing and encouraging them to perform exercises or movements that they otherwise would not do [17], and to follow a specific plan of activities and challenges in the context of their care plans. During COVID-19 distancing conditions, indoor virtual reality exergames show positive results in increasing the physical states of users while maintaining high user satisfaction [18]. In a telerehabilitation exergame platform, which combines mobile devices and remote sensors, stroke patients showed signs of balance improvement [19].

One of the fundamental techniques of the serious game design is gamification. This goal, in the research areas related to this work, is mainly to increase physical and cognitive activities [20]. In addition, with the combination of virtual and augmented reality technologies, user activities can be increased as part of healthcare solutions [21]. It should be noted that despite virtual reality applications showing positive results, they still require additional, and in most cases, expensive hardware [22,23,24]. Another disadvantage of VR is the absence of representations of the user’s body, highlighted by an exergame aimed at reducing risk fall for the elderly [25]. This game promotes AR over VR due to the increased sense of body presence AR provides and the advantage of operating on low-cost commodity hardware.

Research implementations based on the Microsoft Kinect system, a hardware extension to the Microsoft XBOX using RGB and depth cameras, show that such platforms can be highly efficient in medical aspects and support exergames [26,27,28]. Nintendo Wii sports is also a platform that has similar, positive results [29]. There are AR examples that increase the motivation and monitoring of users while exercising and help them maintain their physical conditions [30]. Pokemon Go is a widely used AR game that motivates the user to go out and search for dropped virtual objects [31,32]. Additional solutions refer to mobile AR exergame applications that use virtual objects and spatial audio to encourage player movement [33]; some focus on specific user groups, e.g., motivating children to move by finding and defusing virtual calorie bombs in a real world environment [34]. Applications using the Google ARCore framework can be found in the context of in-house navigation systems [35,36] and focus on the aspects of improving the users’ quality of life.

The use of IoT wearable devices, which are capable of monitoring the users’ physical state [37], is also beneficial to achieve the goals of these games. IoT wearables in medicine are small and flexible electronic devices that can measure physical status and record physiological parameters of the user [38]. Smart wearables are used for fitness [39] tracking, recommendations, and guidance, and can record physical activity aspects [40] along with other measurements, such as heart rate or blood oxygen saturation. Such devices are the core hardware components that realize quantified-self concepts, which refers to the engagement of individuals to self-track any kind of biological or physical information [41]. Rehabilitation data recorded by IoT devices can be stored in the cloud. Related research in the concept of sensor data analysis [42] indicates that these data can be saved, queried efficiently, and accessed by web interfaces.

Combinations of AR and IoT can be found in cases of communication between mobile devices and static infrastructure. Such examples can be found in agriculture, where they facilitate in improving precision farming, especially for indoor planting [43]. In smart cities, IoT sensors provide location-based information presented in the user interfaces, as virtual objects through their mobile phones. This information can vary from transportation announcements [44] to general context defined by dynamically-placed markers [45]. Real-time environmental data are demonstrated by AR mobile applications providing engaging ways of IoT sensor data visualization [46].

Studies that indicate a gap between AR and IoT in the healthcare context discuss architecture and technologies that can be used to perform such integration [47]. Intelligent assistive systems, combining Internet of Things, augmented reality, and adaptive fuzzy decision-making methods help users complete their daily life activities [48]. IoT solutions using the Simblee platform for health data monitoring aim to improve patient engagement during physical rehabilitation with the use of AR [49] for iOS devices.

Although studies show examples of combining AR exergames with IoT, the linking of these two as a prototype exergame for mobile commodity devices with hardware independence, remote configuration, web interface, cloud functionality, and IoT health monitoring, is the key contribution of this work.

## 3. Materials and Methods

### 3.1. System Overview

This prototype is a form of an interactive serious game quest which, by using AR, displays objects in a local home environment and challenges the users to interact with them. At the same time, users’ biometric data are logged for local and remote monitoring. These data are recorded with both built-in mobile device sensors and with the use of IoT wearable devices, such as smartwatches. A web interface is also introduced, which has the role of supervising the game sessions and configures the gameplay parameters and scenarios. Using this interface, a corresponding healthcare professional can monitor the progress of the users, reconfigure the game parameters, and analyze the game and health monitoring results. An overview of the approach is presented in Figure 1.

The system consists of three core components, which are analyzed in the following sections: (a) the *AR game development framework* that runs on mobile phones, (b) the *smartwatch app* that can be installed on WearOS smartwatches, and (c) the *cloud platform and web app* that provides remote game processing, user vital signs monitoring, and results visualization.

#### 3.1.1. AR Game Development Framework—Mobile Application

The main functionalities of the game platform are implemented in the mobile application. The game displays to the user’s screen the different quests and opens up a camera window, showing a view of the area in front. Using AR, the game options and actions are placed on different surfaces, asking the user to walk and interact with the virtual objects, as part of predefined game challenges. The AR functionality, regarding the process of displaying and selecting answers, is implemented using the Google ARCore framework (https://developers.google.com/ar (accessed on 28 March 2022)). Although this framework was preferred due to the seamless integration with the Android ecosystem, there are other AR libraries that can achieve similar results. Apple’s ARKit (https://developer.apple.com/augmented-reality (accessed on 28 March 2022)) is a mobile framework that utilizes AR elements and presents similar results. However developing ARKit requires not only Apple test devices but also Apple computer infrastructure and special licenses, which lead to limitations, making development less flexible. AR.js (https://github.com/AR-js-org/AR.js (accessed on 28 March 2022)) is a lightweight library for AR on the web, which can be used in mobile devices by opening up a web browser window. Amazon Summerian (https://aws.amazon.com/sumerian (accessed on 28 March 2022)) and HoloKit (https://holokit.io (accessed on 28 March 2022) ) are AR libraries that offer mobile device support, but their core AR elements when reaching the native device level are provided by ARCore and ARKit, depending on the platform.

All game quests, their types, and levels, can be selected by the user before the game or can be pre-configured by the healthcare professional through the web application in the context of a personalized care plan. Besides the game functionalities, the mobile app is also responsible for recording and analyzing the measurements supported by the system using the device’s built-in sensors. These measurements mainly refer to hand trembling and user movement (exploiting the raw data from the accelerometer sensors), and after each successful or unsuccessful test, the score is calculated along with recorded health measurements. These results are shared to the professional’s web app via the cloud.

#### 3.1.2. Smartwatch Application

A core feature of the system is integration with wearables for capturing activity and biosignal measurements. The features are implemented through a smartwatch application that can be installed on the device that is paired with the smartphone. The app is launched automatically when a game session starts, and during the game, the smartwatch and the mobile applications communicate continuously using Bluetooth technology. The smartwatch records the heart rate, user movements, and steps, and relays the data to the mobile app, while at the same time, it indicates the results live on the watch screen.

#### 3.1.3. Cloud Platform and Web Application

The web application is mainly an administrative environment that offers to the healthcare supervisor game customization options along with user result visualization. The end user can obtain a view of his/her game history and his/her rankings in the game leaderboards. The supervisor obtains a list of the user’s game sessions and has access to game results, such as correct answers or mistakes, the score of each game, and session gameplay duration. Health-related data, such as steps, heart rate, and hand trembling, are also presented with visual lines and pie charts. These data can be classified and assessed, providing valuable information to the persons that are monitoring the users, or the users themselves. The supervisor has the ability to set the content, the types, and the difficulty of the various challenges for user tests, either generically for all games ot by personalizing the game of specific users. The data exchange between the mobile and web applications is performed through a serverless cloud platform. This set of microservices simplifies the deployment and instantiation of the platform and the game elements, ensuring the overall security, modularity, and scalability, and facilitating the system monitoring for smooth operations. It is responsible for storing game configuration, results, measurements, and for handling communication between web and mobile applications, acting as a midpoint.

### 3.2. The Augmented Reality Features

The gamification experience is achieved through integration of Sceneform library-a 3D framework with a physically-based renderer that is optimized for mobile devices and and makes it easy for one to build augmented reality apps without requiring OpenGL. (https://github.com/google-ar/sceneform-android-sdk (accessed on 28 March 2022)). Sceneform provides ARCore functionality with Android interfaces and classes through an API. The ARCore utilizes the device’s hardware, and by exploiting the library capabilities, offers an AR experience to the users. The UI of the game is presented through a fragment widget, which is provided by the library and is called an *AR fragment*. In this fragment, the view of the camera is rendered along with additional 3D objects that are generated by the AR and are called *anchors*. An anchor can generate its 3D model from an asset file or create a dynamic shape (rectangle, triangle, cube). In order for the anchors to remain steady in their positions, the *plane* app elements are used. Planes are basically surfaces within the scenes that allow the anchors, which are generated in this area, to remain motionless. The planes are invisible on the camera and are used only for classification and recognition. All of these UI elements, which are tailored in a Sceneform view and are initialized with AR fragments, consist of *AR sessions*.

Callbacks for the environment changes and for the detection of anchors and planes are provided along with frame update listeners. When the session starts, the device records the camera input, recognizes objects and items that are in front, and classifies surfaces in order to generate planes. The session also stores information about all plane positions and provides functions to insert, update, or delete planes. The distance between a plane and the device can be calculated as well as the distance between two random planes. These parameters are exploited during game runtimes to create smooth experiences for the users and implement intuitive game controls as part of the real world environment.

### 3.3. The Game Platform

The prototype of the proposed solution includes a framework for the development of serious AR games, quests, and challenges exploiting ready-to use game elements and functionalities. Each game consists of multiple series of questions with different types and difficulty levels, and for each level, different metrics and configurations are applied in order to realize the gamification concept for the users. The overall game configuration, which includes the various levels, challenges, and all operational parameters, can be defined through the web application from the healthcare professionals. It should be noted that these parameters can be personalized for each user and adapted during the runtime in order to tailor the overall solution and its operation to the specific usage scenario and the care plan requirements. At the same time, the use of metrics and configurations allows professionals to assess different parameters of users’ physical and mental statuses, and to effectively motivate them towards specific activities in the context of a personalized care plan. The sets of questions that will be presented constitute a session and are based on the following types:*Text*: a text question is presented on the top of the screen and the answers are displayed in the AR Scene. These answers can be numbers, such as the result of a mathematical expression or text, for knowledge-based scenarios.*Object*: this question type throws three objects (e.g., animals, things, balls) in the scene and asks the user to identify and pick the correct animal that is requested.*Box*: in this current question type, three boxes are presented in the area around the user. These boxes contain hidden answers and the user has to walk through them in order for the answer to be revealed.*Sort*: a number of answers are thrown. These answers have ascending connections between each other and the user is called to put them into order. Usually the style of this question refers to sorting numbers, days of week, months, etc.

### 3.4. Implementation

The system is divided into three applications and the implementation of each application consists of a set of different components, which realize the required functionalities. One of the main goals for the overall system architecture is to address the requirements for modularity and scalability to ensure possible future extensions. Therefore, the platform’s components are designed based on the maximum possible simplicity, with clearly defined interfaces and abstraction layers, while maintaining performance and efficiency at high levels. The development tools and languages chosen and the way they have been used increase interoperability with third party services and allow future integration with other platforms.

The components of the system that operate in the context of the mobile application are: (a) the *menu manager* and (b) the *game controller*. Wearable applications contain the (c) *steps manager* and (d) *heart rate manager* components, which take care of recording and broadcasting measurements to the mobile. Web applications include (e) the *web results manager*, which visualizes the results of the past game sessions, and (f) the *web game manager*, which sets the parameters for the games. The cloud platform component consists of storage, authentication, and synchronization controllers. Storage controllers are responsible for storing (g) health data, (h) game scenarios, and (i) game results while (ia) authentication and (ib) synchronization controllers handle the user login status and data syncing accordingly. A complete component diagram is presented in Figure 2.

Mobile and wearable applications are designed following the Android activity architecture, i.e., a set of self-contained screens that include all functionalities that they require and exchange messages with each other. The smartwatch app consists only of one activity that handles the sensor readings, displays the metrics on the screen, and broadcasts them to the mobile app. The mobile app, on the other hand, contains various activities that correspond to the (a) *menu manager*, (b) *results manager*, and (c) *game controller*. The web application is a single page app, which was developed using the Flutter framework (Flutter: (accessed on 28 March 2022) build apps for any screen—https://flutter.dev). The backend operations are based on Firebase cloud services, especially the Firebase Realtime database (Firebase (accessed on 28 March 2022) Realtime database: https://firebase.google.com/docs/database (accessed on 28 March 2022)). Through the database, the connection between the mobile and web apps is implemented, as well as the synchronisation and storage of all system data.

The mobile application components (a, b) handle the system flow and manage the connections with the other applications. The menu manager (a) component consists of the *results manager* and the *game manager* modules. Results manager presents the results data of the games in grids, graphs, and tables, and these data are also stored to the cloud for remote access using the web app. Game manager sets the level and type of the game and sends the start command to the game controller. The game controller (b) component consists of the following modules: (a) the *measurements controller*, (b) the *ARCore controller*, (c) the *answer validator*, and (d) the *session controller*. The measurements controller calculates trembling, steps, and handles the connection with the smartwatch application. The ARCore controller manipulates the ARCore and Sceneform libraries and provides the modeled outputs to the game controller component. The session controller manages the operation of the game: it receives the start command, obtains the correct assets for the answers, initializes the ARCore controller, reads the measurements from the measurements controller, presents the game and the other elements on the screen, and finally handles the user interactions.

The smartwatch application components (c, d) communicate with the watch sensors and send the measurements to the mobile application via Bluetooth. On a technical level, the reading of the sensors and the communications with mobile are feasible with the use of the integrated Wear OS API.

The web application components (e, f) implement the workflow of the web interface. Initially, the web results manager (e) obtains the data stored by the mobile results manager component in the cloud and presents them to the supervisor. This element also undertakes the operations of setting the parameters, types, questions, and levels for upcoming games. Users can access their results in the results manager activity and view their progress, points, scores, performance etc. The users also have the option to share the results with their healthcare specialists or family and friends.

## 4. Evaluation and Results

### 4.1. System Operation

The game session begins when the menu manager provides the base navigation of the app and also sets the preferred difficulty level and quest type options. This configuration is sent to the game controller in order to initiate the session in the app UI by presenting the AR activity. Based on the quest type, the appropriate questions and possible required assets are loaded to the memory. The measurements controller is initialized and establishes the connection with the smartwatch app (if the watch is paired with the phone). It also initializes the phone’s accelerometer sensor, for detecting hand trembling, and a step counter sensor that is going to be used if the watch’s sensor is not available. The game iterates through all the different challenges of the quest. A challenge (e.g., a question) is displayed as text at the top of the screen and the answers are rendered on the AR fragment. Each answer is modeled as an anchor that is placed on a plane that is identified in the area around the user.

In order to avoid conflicts between anchors, we introduced an anchor placement algorithm based on experimentation in real user environments, such as living rooms, offices, gardens and kitchens. Figure 3 illustrates the flow of the algorithm, which is executed for each question. The answers of a question along with their data (labels, assets, etc.) are stored in a list in the system memory while the AR session is initialized and plane detection starts. For each answer, the algorithm first obtains a random possible location on one of the detected planes. Then, it checks whether this location fits the following criteria:Distance from camera is greater than 2 m;Distance from camera is less than 7 m;Distance from the nearest anchor is greater than 2 m;

**Figure 3 sensors-22-03181-f003:**
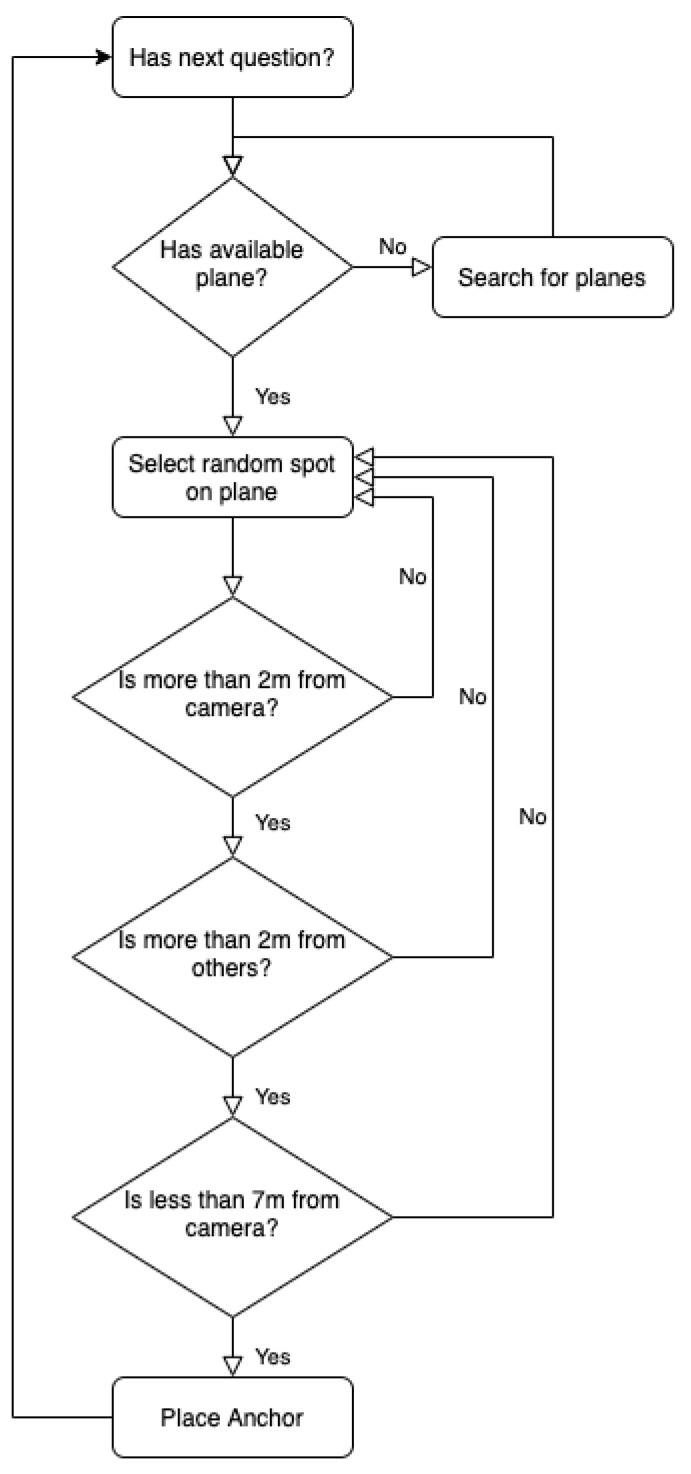
Flow diagram.

If these constraints are met, the anchor of the answer is placed and the corresponding 3D model is benign rendered.

When the user taps on an answer, this answer is stored and the whole iteration proceeds to the next question. In addition to storing the result of the question, metrics that were recorded during the session are stored. These metrics include:*Duration*: the time that the user needed to find, catch, decline, or accept the answer.*Miss hits*: the failed attempts of catching an answer (due to long distance from the object or failure in aiming at the object).*Incomplete sessions*: sessions that are terminated by the user before finishing.*Steps*: the steps that are counted either by the watch or by the phone sensor.*Heart rate*: full heart rate series as recorded by the watch, along with min/max/avg indications.*Hand trembling*: using the accelerometer sensor of the phone, the hand trembling can be detected and classified either in a waveform format or in classification categories, based on the accelerations that are detected from the sensor.*FPS and battery drain*: FPS are calculated using the frame update listener of the AR fragment and exported in min/max/average. Battery drain is the delta of the current battery’s percentage.

### 4.2. Gameplay

The game offers an AR experience to the user combined with real time steps, heart rate, and trembling indicators. After picking a set of questions, level, and type (can be pre-defined for the user by his doctor), the AR game session starts. The goal of the game is to walk around the area in front and try to catch the virtual objects presented in the screen. Types of the AR objects are defined by the question type of the current game quest. There are currently four game quests supported. The gameplay for each question type that is currently supported among with snapshots of the IoT wearable app is shown in Figure 4. The device that this example was running on was Huawei P20 Lite-ANE LX1 ( CPU: 2.36 GHz, RAM: 4 GB, 16 MP Camera).

Type (a) refers to questions that display a 3D model as the object of the quest and the user is asked to identify the requested object and tap on it. Answers presented in free text are displayed as depicted in (b). Type (c) refers to box-type questions, where the user has to tap on the box for the possible answer to be revealed. Finally, in type (d), the user has to pick the displayed answers in a certain order. The positive effect of the gameplay is that the user will have to walk around in order to catch the answers and continue in the game. The context of questions can be defined based on general interests of the user, resulting in an exergame that increases his/her physical activity and also keeps the user mentally motivated by being close to his/her interests.

### 4.3. Cloud Platform and Web Application

The core part of the system is the cloud platform for the storage of the game scenarios and the results, as well as the web app, as the platforms endpoint for healthcare professionals. Figure 5, Figure 6 and Figure 7 illustrate the main functionalities of the web app. The app focuses on the visualization of the results and the rankings, which are presented, based on a point system that classifies the users’ results in tiers. In the prototype implementation of the game scenario, three tiers are proposed; namely silver, gold, diamond, and challenger. Heart rate and trembling are presented in the answer’s screen following the format of line and pie charts The rest of the metrics are also presented in the dashboard.

### 4.4. System Performance

The developed prototype mobile application demonstrates an AR exergame that has the advantage of not requiring any extra hardware. The prototype was tested 10 times for each question type on four devices: Galaxy A20, Galaxy A7 2018, TCL 20 5G, Huawei P20 Lite.

The following tables show average performance results of 10 sessions for each type. Each session contains 10 quests. The battery drain is calculated by the difference of the battery percentage between the start and the finish of each session. The FPS are counted from the first to the last test. Table 1 shows the average performance metrics from 10 tests of text type questions for each device. Table 2, Table 3 and Table 4 show the same metrics from object, box, and sort challenges, accordingly.

Results show that the adoption of box type questions results in smoother gameplay due to the lack of HD textures or multi-edge polygons. The most resource hungry question type is object, due to the rendering of 3D models (assets). Text or sort challenges contain models from Android UI view elements (TextView, ImageView) and they can run smoother than the object questions but not smoother than the box questions. Android UI view elements have better performances than 3D models because of their decreased sizes in edges, polygons, colors, and surfaces. Even with differences between average FPS measurements of each question type, as long as the FPS indication lies around 30, the game is playable and no frame drops can be easily spotted.

### 4.5. System Limitations

Various factors affect the performance of the game, although they do not affect the gameplay to a significant extent. Based on the mobile device model, the lighting conditions and the layout of the scene, screen frame drops are observed if the session lasts unexpectedly a long time. To address this issue, performance optimizations and gameplay suggestions are introduced:Stop the detection of extra planes when all the answers are placed;Prompt to limit the total answers of the questions to a maximum of four;Suggestion to play the game in a large space with plenty of natural light available (e.g., living room or courtyard).

Furthermore, this prototype is only available to Android devices that support Google AR Services ( List of available devices can be found here https://developers.google.com/ar/devices (accessed on 28 March 2022)), although implementation for iOS is concerned about future extensions.

### 4.6. Game Analysis

Apart from the performance analysis, evaluation of the core gamification aspects were also performed. More specifically:The game was tested on 18 users aged 20–40 and 15 users aged 40–60.Each session contains 10 questions; the parameters of steps, session duration (in minutes), heart rate, and hand trembling were recorded.For the steps and heart rate measurements, a smartwatch device was used.Trembling was calculated using the device’s accelerometer and classified on levels from 1 to 4.

The following tables show the results of 10 sessions for each question type. Table 5 shows the average medical metrics of users aged 20–40 and Table 6 the same metrics from users aged 40–60.

Results show that the users are physically active while playing the game, keeping a good pace and with average heart rate. Comparing age groups, results show that the average game duration, steps, and trembling of the 40–60 group age were slightly larger than the metrics of the other group, while heart rate averages followed the opposite direction.

### 4.7. Technology Acceptance Evaluation

The main objective of this work was the development of the AR gamification platform and the proof of concept for combining the AR and IoT technologies in the health domain. Therefore, the evaluation of the system focused on the analysis of the overall architecture and the assessment of its usability. For the evaluation of the usability of the system, a technology acceptance questionnaire (Table 7) was produced and circulated to the previously presented user groups. The questionnaire included a list of questions in order to assess the *perceived usefulness*, the *perceived ease of use*, and the *intention to use*, each one of which could be answered with one of the following responses:Totally disagree (−3);Disagree (−2);Slightly disagree (−1);Neither agree not disagree (0);Slightly agree (+1);Agree (+2);Totally agree (+3).

**Table 7 sensors-22-03181-t007:** Technology Acceptance questionnaire responses.

Technology Acceptance Question	Average Result
I think the AR game is a good idea.	2.3
I believe it is easy to use the AR game mobile app.	1.9
I believe it is easy to use the AR game wearable app.	2.0
I think the AR game is a user friendly technology to interact with.	2.1
I did not have any problems using the AR game in my room.	1.3
The use of the AR game will have a positive impact to my health.	1.6
The use of the AR game could help me to become more active.	1.8
I have the intention to use the AR game daily.	1.9

The above results indicate very positive feedback from the users on the AR game system and very good perceptions on the impact that the use of the system had on their health. Some usability aspects could be improved as well as the ability to use the system in smaller rooms and more complex environments.

## 5. Discussion and Conclusions

The proposed solution combines an integrated and modular environment, state-of-the-art technologies, and tools providing benefits both for the end users and healthcare professionals who can monitor and configure the system operation. Unlike other serious games for health, this AR prototype uses technologies that eliminate the need for extra and expensive equipment, such as gaming consoles, external modules, cameras, screens, etc., and in parallel, monitor very important aspects of users’ physical and mental health conditions.

An additional benefit of the mobile application is the mobility and the ability to perform AR sessions in a home environment. This is the result of a software configuration and methodology that was followed, which (continuously) takes into consideration the status of the real-world and the related physical constrains. This approach allows for a design, a user-friendly hybrid experience, which can be used easily by any user, including elders, not only by tech-savvy persons.

Another advantage is the integrated connection with a wearable that monitors the users’ health data, promotes quantified-self, and provides a complete clinical view of the users’ physical statuses while playing. The combination of AR and IoT is not something new; however, it is not common in a prototype for serious games in the healthcare domain. This is one of the unique characteristics of the proposed solution, which provides a ready-to-use environment for AR game development, which integrates IoT sensors by design. Therefore, a holistic solution is created through which users and healthcare professionals can effectively collaborate to apply personalized care plans, improve physical activity, and promote well-being.

Along with cutting-edge technological benefits, core reasons as to why users are motivated to play the game involve its nature and the gameplay experience. The game scenarios and the multiple possible game implementations may attract the interests of users, and as a result, the chance that the game becomes obsolete is considerably reduced. In addition, the fact that game implementation and the design of the respective scenarios and challenges are coordinated by healthcare professionals provide further motivation for using the system, since the users perceive that it is tailored to their particular needs. This motivation is the goal of using the system and results in an increase of the users’ physical activity and improvement of mental status through well-designed game challenges.

The scoring and ranking system of the mobile and web apps also make the game more competitive and intriguing to play. Detailed versions of the results are accessible through the web application, which shows an extensive image of the stats and metrics with charts and pie diagrams. The fact that the healthcare professionals are actively participating in the system, assessing the results of each game and configuring the different operational and gameplay parameters are other advantages. The services that the web application offers allow the healthcare professional to continuously monitor the user while exercising and providing appropriate feedback. Apart from the results monitoring, the web application’s ability to preset game sessions gives the professional the option to not only set the game parameters, but to replay game sessions and check the physical progress of the user. In addition to the activity motivation aspect, this system can be demonstrated as pleasant for the user, via retrieving important health data, which can diagnose a possible disease. The hand tremor raw data measurements, using only the device’s built in accelerometer sensor, can be utilized in diagnosing tremor-based muscle or neural diseases.

The aforementioned aspects are the main novel contributions of this work, which are not currently available as part of a single offering. It should also be noted that the proposed solution not only brings together unique sets of technologies and features for gamification using AR and IoT in the healthcare domain, but also creates a modern development platform through which healthcare professionals are able to create new innovative environments to interact with their patients and monitor their health. Table 8 illustrates a comparison between the proposed solution and the related literature. The columns of the table correspond to our platform’s capabilities, technologies, and characteristics, which are: (a) *AR*: the solution uses augmented reality technologies, (b) *IoT*: the solution uses of Internet of Things technologies, (c) *professionals*: the solution supports the active involvement of healthcare professionals, (d) *physical activity*: the solution promotes an increase of physical activity, (e) *health monitoring*: the solution supports measuring a user’s physical activity, biosignals, or cognitive status, (f) *gamification*: the solution includes gamification characteristics, serious game scenarios, or challenges, (g) *hardware type*: the type of hardware requirements for system operation. Concerning hardware requirements, the following types are identified: *S*—stationary hardware is required (e.g., sensors or cameras on fixed spots of the room), *P*—portable-handheld hardware is used, *C*—indicates that only commodity devices are used and no special equipment is required.

The analysis of the literature and comparison of the different capabilities highlight the novelty of the solution, as a single offering, which can demonstrate substantial impacts in the healthcare domain by facilitating the development of innovative interventions as part of a personalized care plan. The proposed system enables the provision of advanced remote monitoring capabilities with the active participation of healthcare professionals who can change the system parameters during operation and also create new game scenarios. The proof-of-concept prototype implementation validates the synergy of the utilized technologies and the capability of offering the aforementioned functionality with commodity devices.

In summary, this platform utilizes IoT, AR, cloud, and web technologies by proposing a prototype that, aside from being a physical activity exergame, also monitors the user’s current measurements, provides the doctors with real-time health data, and offers them the option to pre-define and remotely control the game. Furthermore, it is hardware independent and may operate on commodity—and, in most cases, already pre-owned—devices, maintaining very low ownership costs. Overall, it can extend the user’s knowledge base and it offers a fascinating way to improve well-being in the general concepts of quantified self-assisted living.

In future extensions of this work, additional parameters will be supported in order to provide a more detailed view of the users’ health and cognitive status to the healthcare professionals. This includes, among others, integration with affective computing frameworks to assess the users’ emotional status [52,53,54]. Other AR and general computer vision engines will be examined and tested in order to determine whether they are capable of handling such use cases. The gameplay will be further improved, allowing for more advanced interactions with the real world, which will be exploited in the development of more complex game challenges and scenarios. In collaboration with healthcare professionals, the challenges will be enhanced to be aligned with the care plan of the users, and rules for the dynamic adaptations of the game parameters, based on the care plan and the monitored results, will be implemented. As part of this extension, existing implementations for recommendations for quantified-self applications will be integrated [55]. Finally, multi-user operations will be supported in the same physical location, as well as remotely, for providing even better user experience and more exciting gameplay.

## Figures and Tables

**Figure 1 sensors-22-03181-f001:**
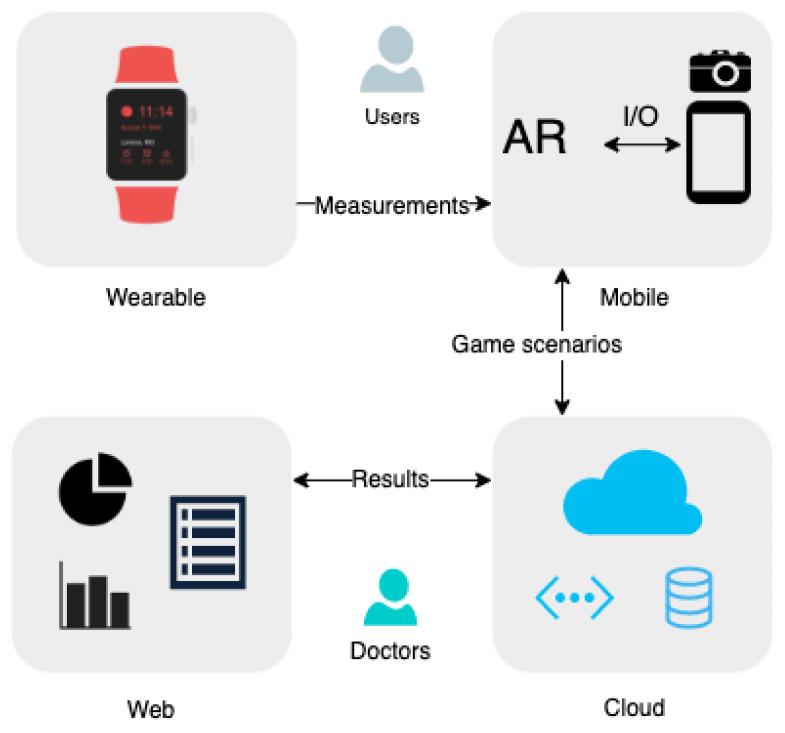
The system overview and the core technologies.

**Figure 2 sensors-22-03181-f002:**
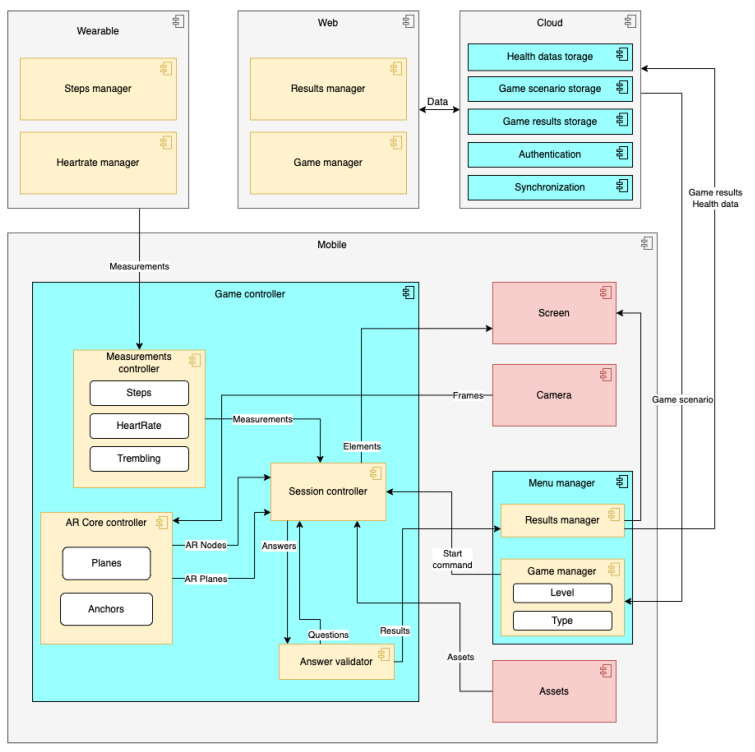
The components of the AR game development platform.

**Figure 4 sensors-22-03181-f004:**
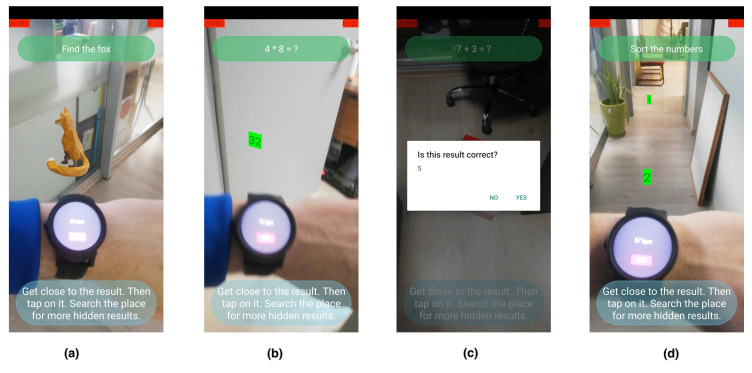
The main game quests: (**a**) object question, (**b**) text question, (**c**) box question, and (**d**) sort challenge.

**Figure 5 sensors-22-03181-f005:**
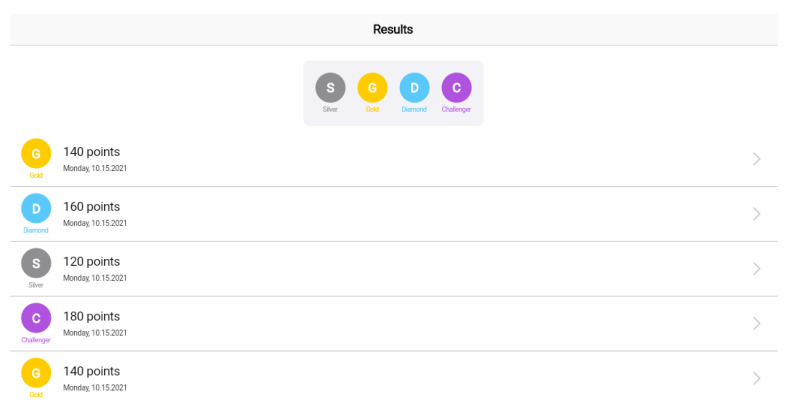
Web app dashboard.

**Figure 6 sensors-22-03181-f006:**
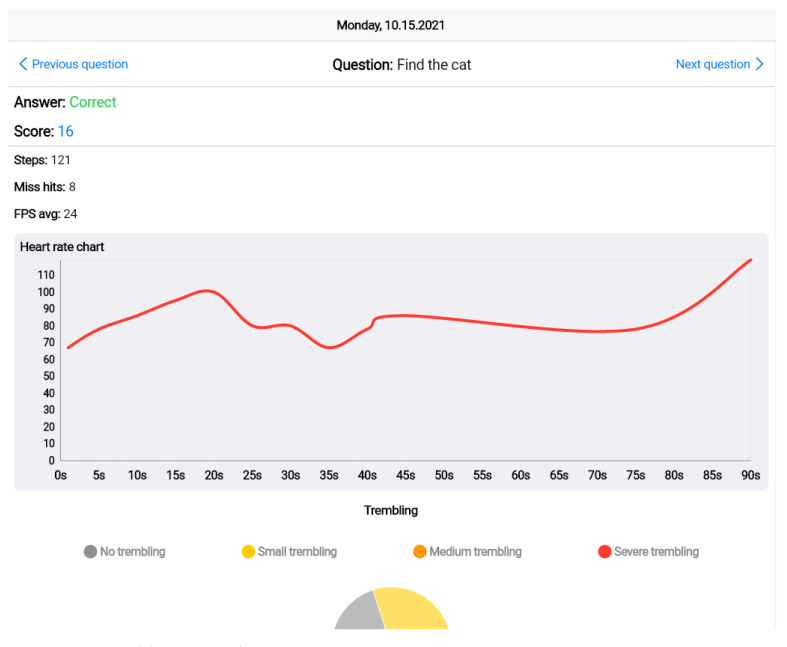
Score and heart rate diagram.

**Figure 7 sensors-22-03181-f007:**
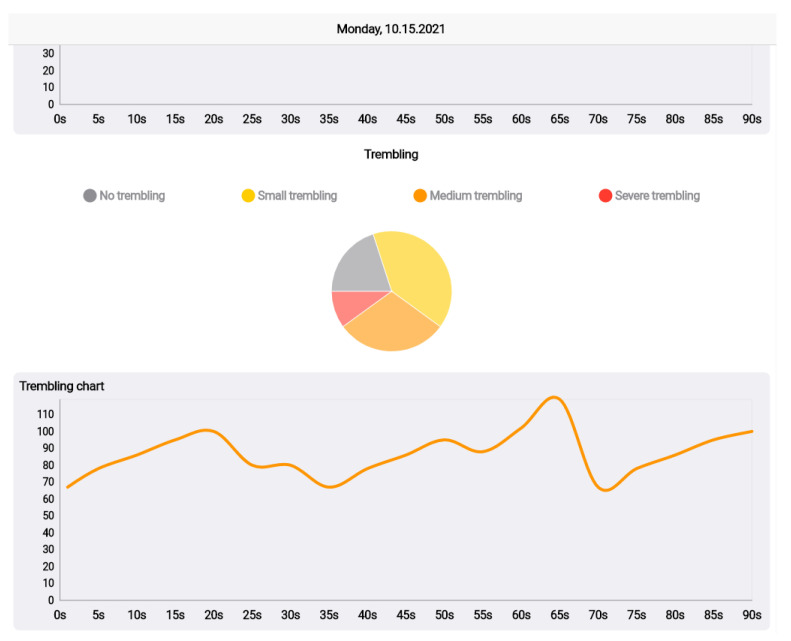
Trembling chart and diagram.

**Table 1 sensors-22-03181-t001:** Average results of 10 tests of 10 text question sessions.

Device	Battery Drain	FPS (min/avg/max)
Galaxy A20	0.9%	17/33/60
Galaxy A7 2018	1.1%	16/32/59
TCL 20 5G	0.6%	16/31/61
Huawei P20 Lite	1.8%	13/29/57

**Table 2 sensors-22-03181-t002:** Average results of 10 tests of 10 object question sessions.

Device	Battery Drain	FPS (min/avg/max)
Galaxy A20	1.1%	11/29/58
Galaxy A7 2018	1.3%	13/31/59
TCL 20 5G	0.9%	14/30/60
Huawei P20 Lite	2.8%	12/27/58

**Table 3 sensors-22-03181-t003:** Average results of 10 tests of 10 box question sessions.

Device	Battery Drain	FPS (min/avg/max)
Galaxy A20	0.9%	20/36/59
Galaxy A7 2018	1.1%	23/37/59
TCL 20 5G	0.7%	21/39/60
Huawei P20 Lite	2.3%	18/31/59

**Table 4 sensors-22-03181-t004:** Average results of 10 tests of 10 sort quest sessions.

Device	Battery Drain	FPS (min/avg/max)
Galaxy A20	1.0%	16/31/59
Galaxy A7 2018	1.4%	19/30/57
TCL 20 5G	1.1%	23/34/59
Huawei P20 Lite	1.9%	14/28/58

**Table 5 sensors-22-03181-t005:** Average results of different challenges in 10 sessions for 18 users aged 20–40.

Question Type	Steps	Duration	Heart rate	Trembling
Text	533	4.9	71/86/121	1
Object	581	5.8	82/101/123	2
Box	499	5.3	80/111/129	1
Sort	551	5.7	73/102/128	1

**Table 6 sensors-22-03181-t006:** Average results of different challenges in 10 sessions for 15 users aged 40–60.

Question Type	Steps	Duration	Heart rate	Trembling
Text	600	6.9	65/83/116	2
Object	591	6.8	61/86/111	3
Box	523	6.2	68/88/112	2
Sort	588	6.9	62/86/110	3

**Table 8 sensors-22-03181-t008:** Comparison of system’s capabilities with the literature.

	AR	IoT	Professionals	Physical Activity	Health Monitoring	Gamification	Hardware Type
Pardos et al. [4]			•		•	•	S
Chen et al. [25]	•					•	S
Chanpimol et al. [28]				•	•	•	S/C
Deutsch et al. [29]						•	S/C
Baranowski et al. [32] Lindeman et al. [33]	•			•		•	P/C
Kim et al. [34]	•			•		•	P/C
Zhang et al. [35]	•	•					P
Henriksen et al. [40]		•		•	•		P/C
Pokric et al. [44] Chaves-Dieguez et al. [45]	•	•					S/C
Nam et al. [50]		•	•	•	•	•	S/C
Marins et al. [51]				•		•	P/C
Koulouris et al. [7]	•			•		•	M/C
**Current prototype**	•	•	•	•	•	•	P/C

## Data Availability

Not applicable.

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
