# Peer review of "An IoT-Enabled Platform for the Assessment of Physical and Mental Activities Utilizing Augmented Reality Exergaming"

_sensors, 2022, doi:10.3390/s22093181_

Round 1

Reviewer 1 Report

This article presents an IoT-enabled platform for the health care domain (assessment of physical and mental activity) through the development of serious games (utilizing augmented reality). From a technical point of view, the article is sound and has some very good contributions. Similarly, the article is nicely articulated and easy to follow. The capabilities of the proposed system have been compared with literature.

However, there are some issues/concerns/suggestions that are required to be addressed. These concerns are given in the following:

  1. Description of the research gap is an important component of the Introduction Section. The article under consideration does not provide a brief overview of state-of-the-art along with its limitations. Therefore, the novelty of the proposed framework can not be judged from the Introduction.

  1. Section 2 of the article provides a good discussion on the combination of AR and IoT. However, the authors claim that the linking of this combination (AR and IoT) with a healthcare aspect is the key contribution of this work. The reviewer has some doubts about this claim. Please consider/discuss the following papers in your literature review before making this claim:
  • Architecting Intelligent Smart Serious Games for Healthcare Applications: A Technical Perspective. Sensors 2022, 22, 810.
  • An Indexing Scheme for Telerehabilitation Big Data” IEEE International Conference on Big Data, pp. 5807-5809, Atlanta, Georgia, USA, December 2020.
  • Towards an intelligent assistive system based on augmented reality and serious games." Entertainment Computing 40 (2022): 100458.
  • "Augmented reality and smart sensors for physical rehabilitation." 2018 International Conference and Exposition on Electrical And Power Engineering (EPE). IEEE, 2018.
  • Augmented Reality Enabled IoT Services for Environmental Monitoring Utilising Serious Gaming Concept. J. Wirel. Mob. Networks Ubiquitous Comput. Dependable Appl.. 2015 Jan;6(1):37-55.

  1. Section 3 presents a very nice overview of the proposed system along with the associated core technologies. Furthermore, various parts/modules of the game development platform (based on augmented reality) have been discussed. The reviewer has no concern in this Section as all the details are clear and nicely presented.

  1. On Page 4, line 320, it is stated that: “Figure 4 illustrates the flow of the algorithm which is executed for each question.” I think the flow diagram, which presents an overall system operation, is presented in Figure 3. Therefore, correction is needed.

  1. The main game quests are presented in Section 4.2. These are: (a) Object question, b) Text question, c) Box question, and d) Sort challenge. However, justification is needed for the selection of these game quests in the context of rehabilitation (physical and mental). Various functionalities of the developed web app are presented in Figures 5, 6 and 7. The presented concepts are clear. The reviewer has no concern in this context.

Author Response

Thank you very much for the feedback.

Concerning the scientific gap, the introduction of the manuscript was updated to clearly highlight that “The use of gamification and AR exergames on commodity devices and across hardware platforms, for providing gameplay location independence, ease of use and remote configuration and health monitoring is a combination of methodologies and technologies which is not present in the current state of research. This work proposes a solution that addresses effectively the above needs using the cutting edge features of IoT, AR for mobile devices, data federation and cloud platforms.”

Furthermore, Section 2 is enriched with additional related literature. 

Figure indicators and descriptions have been updated.

In Section 4.2 physical and mental activity rehabilitation are explained more precisely.

Reviewer 2 Report

The paper "An IoT enabled platform for the assessment of physical and mental activity utilizing augmented reality exergaming" presents a platform that joins augmented reality with Internet of Things. Gamifications aims to help user physical activity and to assess their health and cognitive status by using real-time biosignals. The data was analyzed by healthcare professionals.

- Originality/Novelty: The authors provide a novel platform which mixes augmented reality with Internet of Things. Real-time metrics are displayed for user remote monitoring.

- Significance: The results of the research are interpreted properly, along with tables and figures. 

- Quality of Presentation: 

The article is written appropriately, respecting the logical succession of sections. Data and analyses are presented graphically and inside tables. 

The problem statement, like in Figure 4, should have a bigger font, because the users may encounter some difficulties in reading it.

The numbers to be sorted in figure 4.d are too small. Users may feel some stress and this will appear in the results section.

The hardware details about the computer on which the tests were performed are not mentioned in the paper.

Please enhance image resolution. Some figures are unclear, such as Figure 1.

A comparison between the users aged 20-40 and those aged 40-60 should be done in terns of questions, steps, duration, heartrate and trembling values.

As future work, how do you think that a better accuracy can be reached?

- Scientific Soundness: 
The findings and their implications should be discussed in the broadest context possible and limitations of the work should be also further highlighted.

The conclusions and discussions section should be enriched to provide more details about the contribution of their study to existing literature.

Plagiarism detection could not be performed because all pages appear as images. 

- Interest to the Readers: The conclusions would surely interest the readers of the Sensors journal, and not only them, as augmented reality and Internet of Things are in trends. The paper is interesting and will attract many researchers.  

- Overall Merit: The described solution can be implemented by other researchers and can be involved in other activity recognition experiments. 

- English Level: The level of English language is advanced. Through the entire paper, the language was appropriate and understandable, being easy to follow the flow since the beginning.

Author Response

Thank you very much for your response comments.

Concerning the visuals, the diagram’s resolution and fonts are increased following the suggestions. The text size of the questions has been enlarged and better screenshots have been taken.

Detailed information of the hardware that was used in the demo is also provided.

Game and metric results from age groups are now discussed and findings are described precisely in section 4.6.

A new section containing the system’s limitations is included in the manuscript. This section describes in detail the optimizations which have been applied to the solution to overcome the long session minor lag related issues on older devices. This section also addresses the device limitations of the Google AR services.

Concerning comparison with existing literature, new references have been added and compared with the proposed solution in both the Introduction and Related Work sections. 

The Discussion and Conclusions section has been also enriched by this comparison, following the review comments.

Round 2

Reviewer 1 Report

The raised concerns have been addressed.

The article is suitable for publication in its current form.